# The Elusive B Cell Antigen in Multiple Sclerosis: Time to Rethink CNS B Cell Functions

**DOI:** 10.3390/ijms262110771

**Published:** 2025-11-05

**Authors:** Florian Mailaender, Nicole Vasilenko, Maria P. Tieck, Sonja Schembecker, Markus C. Kowarik

**Affiliations:** 1Hertie-Institute for Clinical Brain Research, Eberhard-Karls University of Tübingen, 72076 Tübingen, Germany; florian.mailaender@med.uni-tuebingen.de (F.M.); nicole.vasilenko@med.uni-tuebingen.de (N.V.); maria.tieck-fernandez@med.uni-tuebingen.de (M.P.T.); sonja.schembecker@med.uni-tuebingen.de (S.S.); 2Department of Neurology & Stroke, Eberhard-Karls University of Tübingen, 72076 Tübingen, Germany

**Keywords:** B cells, multiple sclerosis, B cell antigens, oligoclonal bands, antigen presentation

## Abstract

Although the pivotal role of B cells in the pathogenesis of multiple sclerosis (MS) is well established, their precise functions in disease mechanisms remain incompletely understood. For decades, MS-related B cell research has focused on identifying B cell antigens that could induce pathogenic antibodies contributing to the initiation and maintenance of CNS lesion formation and inflammation. In this review, we conducted a systematic literature search to compile and critically assess proposed antigens with respect to their specificity for MS and the plausibility of findings across different publications. We identified 26 antigens in total. Among these, 15 antigens did not demonstrate high specificity for MS, 9 antigens yielded controversial or contradictory results, and 2 antigens may still be regarded as provisional at the time of our analysis. Based on these findings, a primarily antibody-mediated mechanism driving initial lesion formation is not supported by current evidence, although it cannot be excluded entirely. Instead, a secondary immune response to CNS tissue damage—characterized by local antibody production and alternative B cell functions such as antigen presentation, cytokine secretion and cell-to-cell communication—appears more plausible. Taken together, our review highlights the necessity of expanding B cell–oriented MS research beyond antibody production to include a broader spectrum of B cell functions.

## 1. Introduction

### 1.1. Involvement of B Cells in the Pathogenesis of MS

The pathogenesis of multiple sclerosis (MS) has undergone a profound conceptual evolution, with B cells being recognized as central players in both acute inflammatory and compartmentalized disease processes. Whereas T cells have been considered the primary mediators of MS, accumulating evidence from clinical and laboratory studies has firmly established that B cells contribute to disease mechanisms through a variety of unique immune functions [1,2,3].

Key observations supporting the pivotal role of B cells in MS include several lines of evidence. Increased numbers of clonally expanded B cells within the cerebrospinal fluid (CSF) of MS patients highlight their migration and expansion in the central nervous system (CNS) [4,5,6]. Furthermore, the presence of oligoclonal bands (OCBs), synthesized by clonally expanded CSF B cells, remains one of the most characteristic immunological hallmarks of MS, reflecting intrathecal antibody production [5,7,8,9]. Neuropathological studies have identified ectopic lymphoid aggregates with B cells resembling follicular structures in the leptomeninges of some MS patients; these are frequently observed in proximity to areas of cortical demyelination [7,10,11,12]. Immunohistochemical analysis of MS brain tissue also revealed substantial B cell infiltration within both active and chronic lesions. These B cells not only act as antibody producers but also participate in antigen presentation and cytokine secretion, contributing to lesion formation and progression [5,6]. Finally, the robust clinical effectiveness of B cell-depleting agents such as rituximab, ocrelizumab, ofatumumab, and ublituximab in reducing relapse rates, lesion accumulation, and disability progression in MS patients provides further compelling evidence for the central role of B cells in MS pathogenesis [13,14,15,16,17].

### 1.2. Functions of B Cells

B cells are essential components of the adaptive immune system, with multifaceted roles that extend beyond their classical function in antibody production, where they neutralize pathogens and facilitate their clearance [18]. In addition to generating antibodies, B cells influence immune regulation through secretion of a diverse range of cytokines that enable them to promote inflammatory responses or foster immune tolerance [19,20]. Their capacity to interact with various immune cell types—including T cells, dendritic cells, and innate lymphoid cells—allows them to orchestrate immune reactions by delivering co-stimulatory signals and modulating cytokine networks [20].

A particularly distinguishing feature of B cells is their exceptional ability to present antigens with high specificity on MHC class II molecules. Through their B cell receptor (BCR), B cells selectively bind, internalize, and process antigens recognized with high affinity [21,22]. This BCR-mediated mechanism ensures that B cells efficiently uptake their cognate antigens, route them to specialized intracellular compartments, and present resulting peptides on MHC class II molecules to CD4+ T cells [21,22,23]. Such targeted antigen presentation not only initiates potent activation of T helper cells but also shapes T cell memory and immune outcomes [23,24]. Recent research highlights the ability of B cells to focus antigen presentation toward specifically acquired targets, a feature that distinguishes them from other antigen-presenting cells and underscores their central role in modulating adaptive immunity [22,23,24].

It has not yet been conclusively determined which of these functions plays a decisive role in the pathogenesis of MS. For decades, research has concentrated on identifying MS-specific autoantibodies directed against one or more self-antigens. In this review, we aim to provide an overview of these efforts and discuss possible alternative B cell functions.

### 1.3. Distinctions from Previously Characterized Antibody-Mediated Demyelinating Disorders

Neuromyelitis optica spectrum disorder (NMOSD) and myelin oligodendrocyte glycoprotein antibody-associated disease (MOGAD) are both inflammatory, antibody-driven demyelinating disorders of the central nervous system. While NMOSD is characterized by autoantibodies against aquaporin-4 (AQP4) [25,26] MOGAD shows antibodies against MOG [27]. While historically these entities were often misclassified as variants of MS, they are now recognized as distinct diseases based on differences in their underlying pathophysiology, clinical presentation, imaging features, and response to therapy [28].

A primary distinction between MS and NMOSD/MOGAD is the absence of CSF OCBs in most NMOSD and MOGAD patients, whereas OCBs are found in the majority of MS patients [29,30]. Unlike MS, neither NMOSD nor MOGAD exhibit leptomeningeal or cortical demyelinating infiltrates, which are characteristic pathological hallmarks of MS and contribute to its progressive phase [11,12,31,32]. A progressive course of disease, a well-established feature of MS driven by chronic neurodegeneration and cortical pathology, is not a prominent feature of NMOSD or MOGAD. Both typically follow a relapsing-remitting course characterized by acute, often severe attacks and comparatively less pronounced accumulation of disability between relapses [31,33]. For MOGAD, a monophasic course is also often observed [33]. Treatment responses further underscore these distinctions: Standard MS disease-modifying therapies such as interferons, glatiramer acetate, and natalizumab are generally ineffective in NMOSD and MOGAD—and, importantly, some MS treatments can actually aggravate disease activity [34,35,36]. This therapeutic divergence has dramatic implications for clinical practice, reinforcing the necessity for accurate differentiation.

Besides the detection of OCBs in the CSF of MS patients [5,7] elevated serum immunoglobulin G (IgG) are also observed [37]. Notably, increased intrathecal IgG levels might not fully be explained by local synthesis, as calculations indicate that CSF lymphocytes might account for a small fraction of CSF IgG [37,38]. This suggests a relevant contribution of serum IgG, facilitated in MS by blood–brain barrier (BBB) disruption [39,40]. Autoantibodies against endothelial structures—including galectin-3—have been found at increased levels in MS patients, particularly in progressive forms, but typically appear downstream of disease initiation [41,42,43]. Accordingly, analysis of both serum- and CSF-derived antibodies remains important, as B cell trafficking between the periphery and CNS further supports this comprehensive approach [4,44,45,46,47].

### 1.4. Examining Antibody-Repertoires: Methodological Challenges

Examining antibodies in the laboratory confers several methodological advantages. Antibodies are intrinsically stable biomolecules, enabling their reliable detection and characterization even after prolonged storage or transport [48]. Their intrinsic specificity enables direct recognition and binding to target antigens [49], facilitating the identification of immune responses linked to specific diseases. In vitro analysis also permits detailed epitope mapping and quantitative assays, supporting research into disease mechanisms and potential biomarkers.

Nevertheless, laboratory-based antibody studies are subject to inherent limitations, particularly regarding specificity and translational relevance. Antibodies recognizing linear epitopes may fail to engage native, conformationally folded proteins in vivo, leading to discrepancies between experimental findings and actual pathophysiological processes [50]. Hence, methodological selection must align closely with antigenic structure and biochemical context. While ELISA is well suited for identifying linear epitopes, surface-conformational antigens are more reliably detected via cell-based assays [51].

Reactivity toward linear epitopes can also be examined using an array of complementary analytical approaches. Beyond ELISA, techniques including Western blotting, bead-based immunoassays, two-dimensional immunoblotting, and immunoprecipitation have been commonly employed to characterize such interactions. However, reactivity to a linear epitope does not necessarily imply recognition of the protein’s native conformation. This principle is exemplified by antibodies directed against chloride channel anoctamine 2 (ANO2), which exhibited linear epitope binding in a bead-based assay without corresponding reactivity toward the native structure in a cell-based assay, as discussed in the respective results section [52,53]. The translational and pathological significance of such antibodies remains uncertain, since contactin-2–specific antibodies, for instance, were unable to access their antigenic targets in vivo [54].

Furthermore, multiple sclerosis pathogenesis primarily occurs within the central nervous system, where direct sampling of lesional tissue is rarely feasible and typically necessitates invasive biopsies. CSF analysis offers a less invasive alternative, yet still entails substantial procedural complexity and patient burden. Serum sampling, while convenient and broadly validated for immunological studies, reflects the systemic antibody repertoire, complicating the delineation of disease-specific clones amid a substantial background of non-relevant immunoglobulins [6,10,49]. Consequently, antibody-based studies remain indispensable but must be interpreted with careful consideration of methodological constraints and biological context.

## 2. Methods

To provide a comprehensive overview of potentially pathogenic immune targets recognized by antibodies previously described in the context of MS, a systematic literature search was performed in accordance with the PRISMA 2020 guidelines (Figure 1) [55]. The primary search was conducted in PubMed using the query ‘Antibody target AND “Multiple Sclerosis”’. To improve precision and exclude secondary literature, the additional filter ‘NOT (Review [Publication Type])’ was applied, as the objective of this initial stage was to identify only primary research articles. This restriction was not expected to compromise retrieval of relevant studies. The search covered publications from 1981 to 2025, yielding 1273 records. These were screened in consecutive ten year intervals (1981–1990, 1991–2000, etc.). The initial PubMed screening took place between early June and mid July 2025; no automation or AI tools were used at this stage.

Titles and, where necessary, abstracts were screened to determine study relevance. This process identified 112 publications considered potentially eligible, which were subsequently subjected to full text assessment. All manuscripts were accessible via institutional library resources. During detailed evaluation, several studies were excluded. The most frequent reason for exclusion was the lack of a clearly defined self antigen as the antibody target. Studies that initially described an antigen solely as a T cell target were retained if antibody reactivity against the same antigen had later been examined.

A second exclusion criterion concerned studies that investigated antibodies exclusively as biomarkers of disease activity or progression in MS rather than as functional immunological effectors. These studies are presented separately in the section Autoantibodies as Possible Biomarkers. Five further studies were excluded due to specific methodological or conceptual limitations: in two instances antibody reactivity was only hypothesized without empirical evidence, two relied exclusively on experimental mouse models, and one study was excluded because its cohort size (two patients) was deemed insufficient. None of these studies were subsequently validated in independent MS cohorts, which justified their exclusion.

As the initial search was intentionally restricted to a single database, it was later expanded to enhance completeness. AI assisted research platforms were employed to identify additional relevant literature, specifically Research Rabbit and Academic Perplexity. This extended search was performed between July and August 2025. The final data retrieval from both platforms occurred on 12 August 2025, using the free web version of Research Rabbit and the paid Pro version of Academic Perplexity.

To identify related and complementary publications, the library of already selected studies was uploaded to Research Rabbit, and the “Similar Work” recommendations were systematically analyzed. This process yielded additional immune targets and studies investigating antibody reactivity toward previously reported self antigens. Academic Perplexity was primarily used for manual keyword based searches to locate further literature on established immune targets and to assess whether these self antigens had been reported as antibody targets in other diseases, as such findings would argue against an MS specific pathogenic role.

In total, six novel antigens were identified; for an additional five antigens, literature was found documenting antibody reactivity in other disease contexts. Nine publications reported antibody reactivity to the full panel of identified antigens within MS.

To evaluate potential sources of bias within the included studies, all publications examining autoantibodies against self antigens in MS cohorts were appraised using the Critical Appraisal Tools for Use in JBI Systematic Reviews (Appendix A) [56]. Specifically, studies comparing antibody presence in MS versus control cohorts were evaluated using the JBI Critical Appraisal Checklist for Case–Control Studies (n = 55). Studies were retained only if outcome measurements were equivalent between MS and control groups and appropriate statistical analyses were applied. In most cases, it remained unclear whether the two groups differed solely by MS diagnosis. Matching for sex and age was rarely performed or explicitly reported, which is inherently challenging in cerebrospinal fluid based investigations.

For additional publications that analyzed autoantibodies exclusively within MS cohorts, the JBI Critical Appraisal Checklist for Case Series (n = 2) was utilized. These were included because corresponding case–control studies existed for the same antibodies under investigation.

## 3. Results

Through the described literature review, we identified 26 antigens that have been proposed as disease-relevant antibody targets in multiple sclerosis (Table 1). These candidate antigens were subsequently classified according to their biological functions. When the study first describing a given self-antigen in the context of MS could not be clearly established, the symbol ≤ prior to the publication year indicates this uncertainty. The—at times highly controversial—results were critically evaluated to provide a balanced synthesis of available evidence on autoreactive antibodies in MS.

Supporting evidence summarizes studies that reported significant differences between MS and control cohorts for a given self-antigen, whereas contradictory results lists those that found no significant differences. A more detailed overview including cohort size, effect size and methods of all identified studies that investigated autoantibodies in a multiple sclerosis cohort is provided in Appendix A.

Finally, all self-antigens were categorized as non-specific, controversial, or provisional. Greater weight was given to studies evaluating antibodies against native peptide conformations, which are regarded as having higher translational relevance (see Section 1.4). When specificity was observed exclusively for the native structure but not for the linear epitope, classification followed the results from the native conformation.

An antigen was classified as non-specific if it had also been reported in other diseases or if at least two independent studies demonstrated reactivity in both MS and control cohorts, even when intergroup differences were statistically significant. When these criteria were not fulfilled, the antigen was considered controversial if study results were inconsistent—i.e., at least one study indicated MS-specific reactivity, while another reported either absence or non-selective reactivity. Antigens with MS-specific prevalence below 15% (denoted LF) were likewise categorized as controversial, as were those showing inconsistent findings across isoforms or immunoglobulin chains. The designation provisional applied to antigens identified in a single study reporting MS-specific reactivity in a subset of patients but lacking independent confirmation.

### 3.1. Components of the Myelin Sheath: Proteins and Glycolipids

The loss or destruction of the myelin sheath produced by oligodendrocytes that ensheathes axons is a key feature of multiple sclerosis. Myelin is essential for the rapid, saltatory conduction of nerve impulses and for providing metabolic and structural support to axons. The myelin sheath in the central nervous system is a multilayered membrane, composed of roughly 70–80% lipids and 20–30% proteins [116]. Its major protein constituents include proteolipid protein (PLP) and myelin basic protein (MBP), myelin-associated glycoprotein (MAG), involved in axon–glia interactions; connexin-32 (Cx32) and 2′,3′-cyclic nucleotide 3′-phosphodiesterase (CNPase), which support metabolic and signaling functions; and MOG, a minor component with defined immune relevance [116,117]. Due to the demyelinating nature of MS lesions, the major myelin lipids and proteins have been investigated as potential antigens in MS.

MBP, the second most abundant protein in CNS myelin, is crucial for myelin membrane adhesion and compaction. Early studies implicated MBP in the pathogenesis of MS, as adoptive transfer of MBP-reactive T cells is sufficient to induce experimental autoimmune encephalomyelitis (EAE), the standard animal model of MS [118,119]. However, MBP-specific T cells have been detected in the peripheral blood of both MS patients and healthy individuals [120], indicating limited disease specificity. Likewise, CSF analyses have demonstrated anti-MBP IgG-secreting cells in MS patients but also—albeit at lower frequencies—in other neurological disorders (OND) [60,62,64]. Moreover, recombinant antibodies derived from CSF oligoclonal bands in MS do not bind MBP [65]. Taken together, although MBP is a central immune target in EAE, its relevance as an autoantigen in human MS is questionable, and insights from the EAE model are not necessarily translatable to human disease. Recently, molecular mimicry has received attention as a potential pathogenic mechanism in MS. In particular, one study identified structural mimicry between an epitope of Epstein–Barr virus nuclear antigen 1 (EBNA1) and MBP, with sera from MS patients showing increased antibody reactivity against the EBNA1 epitope compared to healthy controls [63]. Nevertheless, subsequent research into molecular mimicry in MS has mainly focused on other candidate autoantigens.

The most abundant protein within the CNS myelin, PLP, also exhibits encephalitogenic potential, as PLP-reactive T cells are capable of inducing EAE [75]. Nevertheless, PLP-specific T cells have been detected in both MS patients and healthy controls [75]. Likewise, anti-PLP IgG-secreting cells have been identified via ELISPOT in the CSF and blood of MS patients as well as individuals with other neurological disorders albeit at lower frequencies in the latter group [73]. Furthermore, recombinant antibodies derived from CSF oligoclonal bands in MS patients do not bind PLP [65]. However, a study by Owens et al. from 2023 utilized a live-cell assay to demonstrate that IgG antibodies from the CSF of a subset of MS patients (46 out of 80) could bind to conformational PLP expressed on cells—a reactivity entirely absent in patients with other inflammatory or non-inflammatory neurological diseases (0 out of 81) [74]. While these findings are promising, they remain constrained by limited cohort size and the absence of independent validation. Furthermore, the authors propose that such antibodies may contribute to lesion formation in MS; however, they are more likely to represent a broader B-cell–mediated immune activation rather than a true primary autoimmune response directed against PLP. [74].

For CNPase, the third most abundant protein in CNS myelin, the humoral immune response was investigated in patient sera [76]. A significantly elevated antibody reactivity was observed, predominantly of the IgM class, against CNPase in a subset of MS patients (52 out of 70)—a finding absent in HC and patients with systemic lupus erythematosus (SLE), and present only in a minority of patients with OND [76]. A subsequent study employing two-dimensional immunoblotting to survey numerous antigens corroborated these results for IgG, though only for distinct CNPase isoforms detected separately in serum and CSF [77].

MAG is involved in axon–glia interactions and exhibits reduced detectability in MS plaques [121]. Increased frequencies of anti-MAG antibody-secreting cells have been reported in the CSF of a subset of patients [59,60]. Moreover, CSF anti-MAG antibody titers are elevated in MS compared with controls, although their detection generally requires highly sensitive assays, indicating only weak immunoreactivity that likely reflects a secondary response to demyelination [58]. In contrast, elevated anti-MAG IgM titers in serum are predominantly associated with the distinct peripheral demyelinating disorder anti-MAG neuropathy [61], making a direct pathological relevance of anti-MAG antibodies in MS improbable.

MOG possesses encephalitogenic potential, and antibodies directed against MOG are capable of inducing an MS-like disorder in animal models [122,123,124]. Previous studies investigating the prevalence of MOG antibodies in MS patients have yielded conflicting results, with some demonstrating significant differences between MS patients and controls [69,70,71], while others found no such distinction [72]. However, with the recognition of MOGAD as a distinct clinical entity—which may present with symptoms overlapping those of MS—this controversy has largely been resolved [27].

Myelin oligodendrocyte basic protein (MOBP), primarily studied as a T-cell immune target in MS [80,125,126], does not appear to elicit a specific humoral response in MS. In a microarray assay assessing antibody reactivity against a broad spectrum of antigens to differentiate MS from acute disseminated encephalomyelitis (ADEM), the presence of anti-MOBP IgG was more indicative of ADEM than of MS, which parallels findings on MOG antibodies [81].

For the transmembrane protein oligodendrocyte-specific protein (OSP), ELISA detected relatively specific reactivity to a linear cytoplasmic epitope in a subset of relapsing–remitting MS (RRMS) patients [78]. The authors postulate that this epitope exhibits molecular mimicry with viral peptides [78]. Such reactivity was found rarely in patients with primary-progressive (PPMS) and secondary-progressive MS (SPMS), and occurred with similar frequency in other neurological disease controls [78]. Aslam et al. corroborated this observation in 2010, but employed a cell-based assay to assess antibody reactivity against native OSP and found no significant differences between MS patients and controls [79].

Another crucial protein for myelin structure and compaction is cerebellar soluble lectin (CSL), an endogenous carbohydrate-binding protein [66]. While early research showed a high frequency of anti-CSL antibodies in the CSF of individuals with MS—with notable sensitivity and specificity in younger patients—these antibodies are also detected, though less frequently, in other neurological conditions and in older individuals [66,67]. Experimental evidence suggests these antibodies have the potential to disrupt myelination, offering insight into immune-mediated mechanisms that may contribute to demyelination [66]. However, the presence of anti-CSL antibodies is not specific to MS and has, for example, been observed with increased frequency in HIV-seropositive individuals [66,67,68]. Consequently, while studies of anti-CSL antibodies provide valuable perspectives on the complex autoimmune processes underlying MS, they do not currently support CSL as a definitive or singular pathogenic target for this disorder.

Glial cell adhesion molecule (GlialCAM, also known as HepaCAM) orchestrates cell adhesion within the CNS and safeguards white matter architecture. Through protein microarray and phage-display methodologies, GlialCAM was identified as a putative autoantigen via molecular mimicry with EBNA1; the corresponding, phosphorylation-dependent epitope is intracellularly localized [84]. Elevated anti-GlialCAM antibody titers were detected in serum from MS patients relative to controls [84], and reactivity correlated with the HLA DRB1*1501 genotype [85]. Furthermore, the presence of anti-CNS mimic antibodies—including those directed against anoctamin 2 (ANO2), alpha B crystallin (CRYAB), or GlialCAM—was associated with increased MS risk, albeit with highly variable effect sizes between studies [85,86]. Notably, Phage Immunoprecipitation Sequencing rarely demonstrated anti-GlialCAM antibodies among MS patients [88]. Testing by our group using multiplex assays identified elevated antibody titers against specific EBNA1 peptides and ANO2—but not CRYAB or GlialCAM—in MS patients compared to healthy controls; however, the number of MS patients tested was limited [87]. Collectively, these findings highlight an unresolved role for GlialCAM-specific humoral immunity in MS pathogenesis, warranting further elucidation.

Galactocerebroside (GalC), a major myelin glycolipid in Schwann cells and oligodendrocytes, was initially implicated in demyelinating disorders based on its association with neuropathy in rabbits [127,128]. Rostami et al. evaluated human serum and CSF for anti-GalC antibodies using four different assays but found no significant differences between patients with MS, Guillain–Barré syndrome (GBS), chronic inflammatory demyelinating polyneuropathy (CIDP), healthy controls, or individuals with other neurological diseases, arguing against a disease-specific role for GalC [57].

Sulfatide, another glycosphingolipid predominantly also expressed by Schwann cells and oligodendrocytes, contributes to glia–axon interactions [129]. Using ELISA with confirmation by thin-layer chromatogram overlay, Ilyas et al. detected increased CSF antibody reactivity against sulfatide in a subset of MS patients (15/76 positive) compared with those with other neurological diseases (OND; 4/57 positive) [82]. In a subsequent lipid-reactivity screening, Brennan et al. similarly found a predominance of anti-sulfatide antibodies in MS patients’ CSF (60% positive), though a notable proportion of OND patients also tested positive (25%) [83]. Notably, recombinant antibodies generated from the CSF of anti-sulfatide–positive patients failed to bind live CNS myelin [83]. Collectively, these findings make a primary pathogenic or demyelinating role for anti-GalC or anti-sulfatide antibodies unlikely.

### 3.2. Axonal, Cytoskeletal, and Neuronal Antigens

Besides demyelinating lesions in MS, histopathological analyses reveal that MS plaques exhibit varying degrees of axonal injury, ranging from structural alterations to outright axonal loss [130,131]. These axonal damages are a major contributor to irreversible neurological deficits and disease progression [130,131]. For this reason, neuronal proteins have also been investigated for their potential to serve as antigens.

Neurofascins (Nf) are cell adhesion molecules essential for the organization of nodes of Ranvier and the integrity of myelinated axons. The Nf186 isoform is a neuronal protein localized at the nodes of Ranvier, whereas Nf155 is the paranodal oligodendrocyte-specific variant [89]. By using ELISA, Mathey et al. demonstrated that serum antibody titers against an extracellular domain of Nf155 were elevated in patients with chronic progressive MS compared to healthy controls, other neurological diseases, and RRMS; these antibodies also exhibited cross-reactivity with Nf186 [89]. Similarly, Mathew et al. detected antibodies against at least one neurofascin isoform in the majority of MS patients (57 out of 71 positive) by immunoassay, albeit without comparison to a control group [90]. However, these findings contrast with studies employing cell-based assays, which have failed to identify anti-Nf155 antibodies in MS patients [53,91].

Using Western blot, Derfuss et al. identified contactin-2a, a glycoprotein involved in the organization of axonal domains at the nodes of Ranvier, as a potential immune target in MS [54]. Antibodies to contactin-2a were detectable by ELISA in the serum of both MS patients and controls and—in lower frequency—in the CSF of MS patients and individuals with OND. In a cell-based assay, anti-contactin antibodies were subsequently detected only in RRMS patients, but not in healthy controls, PPMS, or SPMS cases, and even then at low prevalence (4/51) [94]. Experimental data further suggest that contactin-2a antibodies are unable to exacerbate EAE in rodents with intact myelin sheaths [54], making a primary pathogenic role of these antibodies in MS unlikely.

Sperm-associated antigen 16 (SPAG16), a protein potentially involved in the microtubule apparatus predominantly expressed in the brain (neurons) and testis, was first identified as a putative immune target by Somers et al. in 2008 [92]. Using phage ELISA, increased CSF antibody reactivity in MS patients against an SPAG16 isoform was detected, compared to healthy and disease controls. Subsequent studies confirmed the presence of SPAG16-specific oligoclonal bands in the CSF of some MS patients (5 of 23), as well as elevated anti-SPAG16 antibody titers in serum relative to controls [93]. However, as positive reactivity was also observed—albeit less frequently—in control groups, SPAG16 cannot be considered a disease-specific immune target in MS [93].

Flotillin-1 and -2 are membrane scaffolding proteins found in neurons in lipid rafts and are highly conserved [95]. Cell-based and immunoprecipitation assays found rare MS-specific anti-flotillin-1/2 antibodies in serum, with little to no overlap with other neurological or autoimmune conditions, but their occurrence is infrequent (~2%), and their contribution to pathogenesis is yet unsubstantiated [53,95].

N-acetylneuraminic acid (Neu5Ac), a form of sialic acid expressed on glycoproteins frequently found in neurons, has also been investigated as a potential immunological target in MS. Glycan microarray analysis demonstrated increased anti-Neu5Ac—and cross-reactive anti- N-glycolylneuraminic acid (Neu5Gc)—antibody titers in the serum of MS patients compared to those with ONDs [96]. However, this finding could not be replicated in CSF samples [96]. Notably, elevated reactivity to Neu5Gc has also been documented in unrelated conditions such as Kawasaki disease [97], further challenging the disease specificity of these antibody responses.

### 3.3. Heat Shock Proteins

Heat shock proteins (HSPs)—a group of highly conserved molecules [104]—are prominently expressed in MS lesions, where they play important roles in cellular stress responses. Histopathological studies show increased expression of small heat shock proteins such as HSP27 and αB-crystallin in astrocytes and oligodendrocytes within active MS plaques. These proteins likely help protect myelin and cells from further damage but may also serve as immune targets contributing to disease progression [101].

Gao et al. found no significant difference in CSF antibody reactivity against HSP60/65 between MS patients (10 out of 18) and those with OND (7 out of 17) [98]. For HSP70, antigen-specific T cells are more frequently observed in the peripheral blood of MS patients than in healthy controls or individuals with mycobacterial tuberculosis [104]. Quintana et al. employed an antigen microarray assay to assess serum antibody reactivity against a wide range of antigens in RRMS patients and healthy controls, reporting elevated responses—primarily of the IgM subtype—against HSP60 and HSP70 in RRMS, while reactivity remained low in PPMS and SPMS [99]. Using a similar approach, they found comparable results in CSF when comparing untreated RRMS and OND patients [100]. Nevertheless, both studies also described a general increase in antibody reactivity to diverse CNS antigens among MS patients [99,100], and increased anti-HSP antibody titers have likewise been reported in other autoimmune diseases [105].

αB-crystallin (CRYAB), another small Hsp, is generally undetectable in the healthy brain but is present within MS lesions as well as in the white matter of other neuropathological conditions [101]. Evidence regarding antibody reactivity to this protein remains contradictory. Van Noort et al. demonstrated that anti-CRYAB antibodies are found in both the serum of MS patients and healthy controls, using ELISA and Western blot targeting the unmodified protein [103]. In the already mentioned study by Lovato et al., anti-CRYAB antibodies were detected in only a very small proportion of MS patients (1 of 18) and OND subjects (1 of 20), and were entirely absent from CSF samples [77]. More recently, molecular mimicry has been suggested between a CRYAB epitope and an EBNA1 epitope. Using a suspension bead array, an increased frequency of antibodies targeting this CRYAB epitope was measured in MS patients compared to healthy controls [102], although this could not be confirmed in our own study [87].

### 3.4. Ion Channels

Autoantibodies against AQP4 and other ion channels, such as VGKC, have been found in autoimmune CNS diseases like neuromyelitis optica spectrum disorder and limbic encephalitis; accordingly, ion channels have also been investigated as potential antibody targets in multiple sclerosis [26,132,133]. However, as outlined in the Introduction, NMOSD exhibits distinct clinical and pathological features compared with MS.

The inward rectifier potassium channel KIR4.1 is primarily expressed in glial cells, where it is essential for maintaining potassium homeostasis and facilitating neurotransmitter uptake [106]. Srivastava et al., employing solid-phase ELISA with recombinant KIR4.1, initially reported a high prevalence of anti-KIR4.1 antibodies in the sera of MS patients (46.9%), whereas such reactivity was absent in healthy controls (0%) and rare in patients with other neurological diseases (0.9%) [106]. The pathogenic potential of these antibodies was further demonstrated in animal models [106]. However, subsequent investigations—using both alternative methodologies such as cell-based assays and, ultimately, the same protocol as Srivastava et al.—failed to replicate these findings [107,108]. Consequently, current evidence does not support KIR4.1 as a primary pathogenic immune target in MS.

ANO2 is a calcium-activated chloride channel expressed, among other sites, in CNS neurons [52]. In bead-based assays, increased antibody reactivity to ANO2 was observed in a subset of MS patients (15.5%) compared to population-based controls (3.2%) [52]. In the same study, the concomitant presence of these antibodies together with elevated EBNA1 IgG titers and the HLA DRB1*15 genotype, established risk factors for MS [134,135], was associated with a markedly increased odds ratio for MS [52]. The antibodies predominantly targeted an epitope that appears to be cytoplasmically localized, for which molecular mimicry with an EBNA1 epitope has also been proposed [52,85,86]. In contrast, Lleixà et al., using a cell-based assay, detected ANO2 antibodies in serum (and CSF) of only one patient (1/50) among 282 cases examined [53].

### 3.5. Metabolic Enzymes and Miscellaneous Antigens

The following heterogeneous antigens have also been investigated as potential antigens in multiple sclerosis and were primarily identified using microarray assays that systematically assessed antibody reactivity against a broad spectrum of self-antigens or by methodologies involving electrophoretic separation of complex protein mixtures prior to immunological screening.

Transaldolase and transketolase are key enzymes of the pentose phosphate pathway, which generates NADPH—an essential cofactor for lipid biosynthesis in the CNS. Particularly during brain development and remyelination, as well as for the reduction in oxidative stress NADPH plays an important role [109,111]. For human transaldolase, early studies suggested molecular mimicry with viral peptides from the human T-cell lymphotropic virus type 1 (HTLV-1) [109]. Elevated antibody titers against transaldolase have been reported by Western blot in both serum and CSF of a subset of MS patients (25/87), whereas such reactivity was absent in OND patients (in both CSF and serum) and in other autoimmune diseases, with only a single case of essential cryoglobulinemia testing positive in serum [109]. This specificity was reproduced by the same laboratory in a subsequent study [110]. Niland et al. also observed increased anti-transaldolase antibody titers by ELISA [111]. In contrast, Lovato et al., using two-dimensional immunoblotting, did not detect anti-transaldolase antibodies in MS or OND patients, with only 1 of 18 MS patients positive in serum and none in the OND group [77]. However, they identified a specific reactivity against transketolase isoform c in both CSF and serum of a subset of MS patients, which was not present in ONDs [77]. These latter findings, however, have yet to be confirmed in further studies.

Recombination signal binding protein for immunoglobulin kappa J region (RBPJ) is a nuclear protein involved in the regulation of transcription across diverse cell types. It was initially identified as a potential immune target through a protein microarray assay [113]. To verify this observation, Querol et al. employed ELISA, detecting a statistically significant difference in CSF reactivity between patients with MS and those with non-inflammatory neurological disorders (NIND), but not between MS and other inflammatory neurological disorders (OIND) [113]. No significant differences were observed in serum between MS patients and healthy controls. Immunocytochemistry demonstrated reactivity of MS CSF with RBPJ-transfected HEK cells in only 4 of 30 cases [113]. The authors concluded that RBPJ is unlikely to play a primary pathogenic role—particularly given that anti-RBPJ antibodies have also been reported in unrelated conditions such as breast cancer [113,114].

Archelos and colleagues identified oligodendrocyte-derived peptides whose coding mRNA sequences do not correspond to annotated human proteins, but whose amino acid composition reveals homology to repetitive Alu retrotransposon elements [112]. Subsequent ELISA analyses demonstrated increased antibody titers against these Alu-like peptides in both CSF and serum from MS patients compared to disease and healthy controls, although this reactivity was not specific for MS [112]. Notably, immune responses to Alu repeat sequences have also been reported in other disease contexts [112].

Coronin-1a is a cytoskeletal and actin-interacting protein mainly expressed in immune cells, aiding cell motility and phagosome dynamics [115,136]. Peptide ELISA identified antibodies to coronin-1a in CSF from RRMS patients more commonly (7 out of 31) than controls (3 out of 42) [115]. Small sample sizes and presence of reactivity in OND make it unclear whether this is a cause or consequence of pathology. Furthermore, such antibodies would be directed against immune cells rather than oligodendrocytes [115].

### 3.6. Autoantibodies as Possible Biomarkers

A range of less well-characterized self-antigens have been identified as immunological targets in MS; the corresponding antibodies have primarily been reported as candidate biomarkers, often in association with specific clinical subtypes of the disease. While a comprehensive analysis of these antibodies lay outside the primary objectives of our systematic literature review, we briefly summarize these findings here to provide a more complete overview.

Autoantibodies directed against transglutaminase 6 (TG6)—an enzyme integral to protein cross-linking—have been detected predominantly in the CSF of individuals with SPMS and PPMS, in contrast to patients with RRMS and other non-MS neurological diseases [137]. Importantly, such TG6-reactive antibodies have also been described in a range of other neurological conditions, indicating that their presence is not specific to MS [137]. Anti-ganglioside antibodies have likewise been identified with increased frequency particularly in progressive courses of MS [138]; however, these antibodies are primarily associated with inflammation of peripheral nerves such as GBS [139]. Elevated titers of antibodies against the immunosuppressive lectin galectin-8 have been reported in RRMS patients relative to healthy controls and again progressive MS forms [140]. Although the presence of anti–galectin-8 antibodies does not appear to be unique to MS, an association with clinical disease worsening has been observed in the RRMS subgroup [140]. In the context of acute clinical activity, antibodies with the potential to disrupt the integrity of the blood–brain barrier are more frequently described in patients experiencing acute MS exacerbations or displaying a chronic progressive disease trajectory. This group includes antibodies that target endothelial cells, microvascular structures, and galectin-3; none, however, demonstrate MS specificity [41,42,43]. Also, anti-ceramide antibodies were elevated compared with both OND patients and healthy controls [141]. Moreover, antibodies against ceramides with distinct N acyl chain compositions—primarily membrane-associated—showed certain patterns in MS subgroups but did not correlate with clinical measures [141]. A distinct subset of RRMS patients presenting with predominant spinal cord involvement and optic nerve attacks have been found to harbor serum antibodies against chloride intracellular channel protein 1 (CLIC1) [142]. These antibodies were not detected in other RRMS subgroups, NMOSD patients, or healthy controls, indicating a potential link with a specific MS clinical phenotype [142].

## 4. Discussion

Given the numerous findings over recent decades and the high efficacy of B cell depletion therapies, an important role of B cells in the pathogenesis of MS is well established. Based on the outlined diverse functions of B cells, the following roles of B cells in MS should be taken into consideration: (I) B cells may produce pathological antibodies that directly initiate and promote tissue damage, similar to antibody-mediated diseases such as NMOSD. Alternatively, the intrathecal B cell response may represent a secondary response following CNS tissue injury, involving (II) local antibody production and (III) potential antigen uptake with presentation to other immune cells. Associated with all functions, B cells can influence lesion formation and the inflammatory environment through their production and secretion of cytokines (Figure 2). In addition, Epstein–Barr virus (EBV), which preferentially persists in memory B cells, may further affect overall B cell functions [87]; however, a detailed analysis of EBV’s impact is beyond the scope of this review.

Due to the presence of B cells in CNS lesions accompanied by antibody and complement deposition, as well as CSF-specific oligoclonal bands, the production of pathogenetically relevant antibodies as a primary trigger for lesion development has long been the focus of MS-related B cell research [38,143]. Such antibody-driven mechanisms are well recognized in NMOSD and MOGAD, which share certain clinical and pathological features with MS and were historically regarded as related subtypes. However, as outlined in the introduction, subsequent advances have elucidated substantial differences among these disorders in both their underlying immunopathology and therapeutic response profiles.

More recently, this potential role of B cells has regained attention in the context of the association between Epstein–Barr virus and MS, with cross-reactive antibodies between EBNA1 and GlialCAM, ANO2, and CRYAB proposed to develop via molecular mimicry [85]. Although there have been controversial studies regarding most of these antigens, the possibility of pathological antibodies against one or more MS-related antigens cannot be completely ruled out but seems unlikely given the extensive research in this field over the past decades. In summary, 26 antigens have been proposed as potential targets in multiple sclerosis. Among these, 15 antigens did not demonstrate high specificity for MS, 9 antigens showed partially highly controversial and/or contradictory results, while 2 antigens might still be considered provisional at the time of our literature research. This analysis demonstrates that despite methodological advancements such as ELISA, Western blotting, immunoprecipitation, cell-based assays, recombinant antibody technology, and immune microarrays, the field continues to face major challenges in demonstrating true pathogenetic relevance and disease specificity.

Many antibody responses are detected at low frequency or are also present in healthy controls and in other neurological disorders, and discrepancies between assay formats (e.g., linear vs. conformational epitopes; serum vs. CSF vs. tissue) further complicate interpretation. A defined, disease-specific primary antibody response that might clearly drive active CNS inflammation therefore remains elusive, and the translation of in vitro observations into in vivo disease mechanisms demands continued rigorous validation and replication in independent cohorts. Given the substantial efforts invested over decades, the summarized findings suggest that B-cell functions beyond the production of primary autoreactive antibodies may play a substantial role in MS.

Such alternative functions could include a secondary, local B cell response to tissue damage—possibly initiated by T-cell-mediated injury. This secondary reaction could include the reaction of B cells against cellular debris with the production of antibodies, which may further amplify inflammatory cascades [144,145]. This possible function does not necessarily conflict with the detection of antibodies against, for example, PLP [74], as myelin components may represent targets resulting from local tissue injury. Furthermore, these B cells could additionally act as antigen-presenting cells within the CSF and/or the periphery. This hypothesis is further supported by several studies on CSF and peripheral blood B cell repertoires that showed bidirectional trafficking of B cells in and out of the CSF [4,44,45,46,47]. Thereby lineage trees between CSF and peripheral blood B cells seem to primarily undergo active diversification in the CSF compartment, with B cells clones also acquiring additional mutations in periphery [4,44,45,46,47]. These findings suggest that CSF B cells encounter antigens during CNS inflammation and then recirculate into the periphery, where they might act as antigen-presenting cells and thus feed and sustain autoimmune circuits. In this context, cytokine secretion and cell-to-cell communication are also likely to be prominent functions of B cells. Although these B cell functions appear highly relevant in light of the absence of clearly defined B cell targets, these proposed mechanisms require further clarification.

To advance the discovery and validation of MS-related antibodies, future work should adopt an integrated, standardized, and longitudinal framework. A logical next step involves methodical cloning and characterization of recombinant antibodies derived from patient CSF B cells, ideally preserving native antigen configuration during screening to avoid artefactual reactivity. Parallel multi-center investigations using harmonized live cell-based assays would permit reproducible comparisons between CSF- and serum-derived immunoglobulins across cohorts. Combining B-cell repertoire sequencing with antigen-discovery approaches—such as phage display or cell-based binding assays—would further refine candidate identification by coupling sequence data with antigen specificity and effector function. Finally, incorporating repeated longitudinal sampling can distinguish transient secondary immune responses from sustained primary antibody generation, thereby improving the interpretability and translational relevance of such findings.

In the present review, the expansion of the literature search to cover autoantibodies in multiple sclerosis was facilitated by the use of AI-based tools. While this approach offers significant advantages in efficiency and scope, it is important to acknowledge the potential for bias that may be introduced by AI-assisted searches. Such bias can stem from algorithmic preferences toward particular journals, publication dates, or citation counts, as well as from selection mechanisms that may underrepresent niche or non-mainstream sources [146,147]. Data and algorithmic limitations, including incomplete coverage or unrecognized cultural perspectives, may also compromise the comprehensiveness and objectivity of the review. Furthermore, AI tools can reinforce confirmation bias by preferentially displaying literature similar to initial queries or prompts, potentially reducing the diversity of scientific viewpoints [146,147].

The major limitation of our literature review lies in the heterogeneous methodologies applied across different studies for the same antigens, which makes comparability between studies challenging and heavily dependent on the scientific validity reported by the authors. Therefore, establishing a clear assessment of the rigor of each study is challenging and cannot be sufficiently achieved within this literature review. Overall, the scientific interpretation appears imbalanced, as published articles reporting the novel identification of a potential antigen tend to appear in high-impact journals, whereas follow-up investigations rarely achieve comparable visibility and impact, resulting in the findings presented here lacking equivalent prominence. This disparity in journal impact factors is detailed for each autoantibody study in MS within Appendix A. Such an observation may also contribute to the large number of antigens classified as controversial in the current literature.

## 5. Conclusions

In summary, the highly controversial findings on MS-specific B cell antigens over recent decades indicate that current evidence does not support the existence of a singular, high-prevalence primary autoantibody responsible for MS pathogenesis. Instead, a secondary immune response to CNS tissue damage, involving local antibody production, antigen presentation, and cell-to-cell communication, appears more plausible. Considering B cell repertoire analyses and trafficking patterns indicating migration of B cells from the CSF into the periphery, peripheral antigen-presenting functions warrant closer investigation. Overall, our review underscores the need to broaden B cell–focused research in MS to encompass alternative functions beyond antibody production.

## Figures and Tables

**Figure 1 ijms-26-10771-f001:**
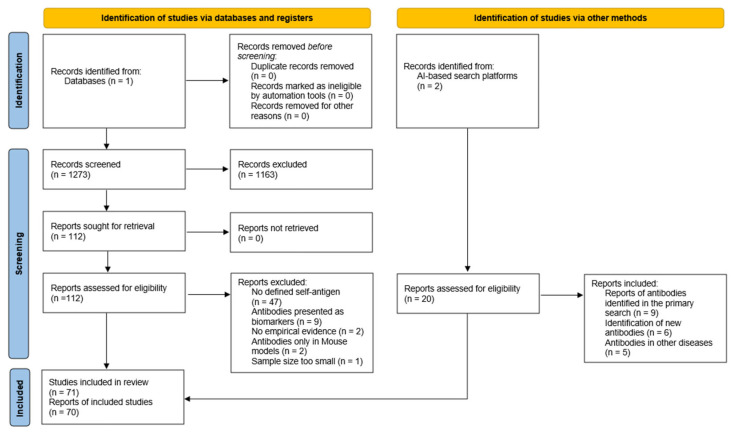
Search Strategy.

**Figure 2 ijms-26-10771-f002:**
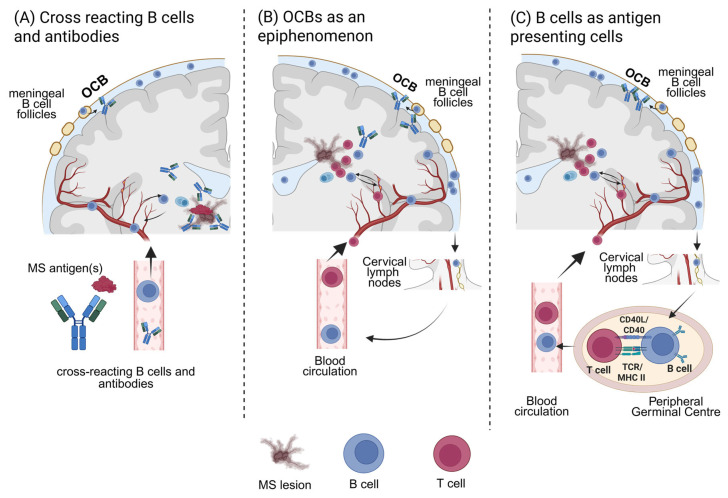
Potential B cell functions in multiple sclerosis. (**A**) B cells may produce pathogenic antibodies that arise via molecular mimicry and directly mediate lesion formation and neuroinflammation. Alternatively, the intrathecal B cell response might reflect a secondary reaction to CNS tissue damage, characterized by (**B**) local antibody production against cellular debris and (**C**) antigen presentation to other immune cells both within the CNS and the periphery. Cytokine secretion modulating the immune response could be associated with all these functions. — Created with BioRender.com.

**Table 1 ijms-26-10771-t001:** Identified Immune Targets.

Location/Function	Self- Antigen	Initial Identification	Method	Molecular Mimicry Reported	Supporting Evidence *	Contradictory Evidence *	Status
Myelin Sheath	GalC	1987 [57]	RIA	No		D/B [57]	Non-specific
	MAG	≤1989 [58]	RIA	No	L/C [59], L/B [60]	L/C [58], R/OD [61]	Non-specific
	MBP	≤1990 [62]	ELISPOT	Yes: EBNA1 [63]	L/P [63], L/B/NS [64]	L/B [62], L/B [60], N/O/CS [65]	Non-specific
	CSL	1990 [66]	Western blot	No	L/C [66], L/C [67]	L/C/OD [68]	Non-specific
	MOG	≤1991 [69]	ELISA	No	L/B [69], N/P [70], N/P [71]	N/O/CS [65], U/B [72], R/OD [27]	Non-specific
	PLP	≤1991 [73]	ELISPOT	No	N/C [74]	L/B [73], L/P [75], N/O/CS [65]	Provisionally
	CNPase	≤1998 [76]	Western blot	No	L/B [76], L/B [77]		Controversial
	OSP	1999 [78]	Western blot	Yes: Common pathogen peptides [78]	L/P [78]	N/B [79]	Controversial
	MOBP	≤2000 [80]	Proliferation assay (T-cells)	No		L/P [81]	Non-specific
	Sulfatide	2003 [82]	ELISA	No	D/C [82], D/C [83]		Non-specific
	GlialCAM	2022 [84]	Microarray	Yes: EBNA1 [85]	L/P [84], L/P [85] L/P [86]	L/P [87], L/P [88]	Controversial
Neuronal	Neurofascin	2007 [89]	Western blot	No	LN/P/SP [89], L/P/CS [90]	LN/P/OD [91], N/B [53]	Non-specific
	SPAG16	2008 [92]	ELISA	No	L/C [92], L/P [93]		Non-specific
	Contactin-2a	2009 [54]	Western blot	No	LN/B/SR+LF [94]	L/B [54]	Controversial
	Flotilin-1/2-Heterocomplex	2017 [95]	Immunoprecipitation	No	N/P/LF [95], N/B/LF [53]		Controversial
	Neu5Ac	2020 [96]	Microarray	Yes: Neu5Gc [96]	D/P [96]	D/C [96], D/P/OD [97]	Non-specific
Heat Shock Proteins	HSP60	≤1994 [98]	Western blot	No	L/P/SR [99], L/C/SR [100]	L/B [98]	Non-specific
	CRYAB	1995 [101]	Western blot	Yes: EBNA1 [102]	L/P [102]	L/P [103], L/B [77]	Controversial
	HSP70	≤1996 [104]	Proliferation assay (T-cells)	No	L/P/SR [99], L/C/SR [100]	L/P/OD [105]	Non-specific
Ion Channels	Kir4.1	2012 [106]	ELISA	No	N/P [106]	N/B [107], N/P [108]	Controversial
	ANO2	2016 [52]	Bead-Based Antigen Array	Yes: EBNA1 [85]	L/P [52], L/P [86]	N/B [53]	Controversial
Miscellaneous	Transaldolase	1994 [109]	Western blot	Yes: HTLV-1 Peptides [109]	L/P [109], L/B [110], L/P [111]	L/B [77]	Controversial
	Alu-peptides	1998 [112]	Phage display	No	L/B [112]		Non-specific
	Transketolase	2008 [77]	2D-Immunoblotting	No	L/B [77]		Provisionally
	RBPJ	2013 [113]	Microarray	No	LN/C/LF [113]	L/P/OD [114]	Non-specific
	Coronin-1a	2013 [115]	SDS-PAGE/ Mass Spectrometry	No	L/C [115]		Non-specific

* Abbreviations in the columns ‘Supporting Evidence’ and ‘Contradictory Evidence’: The first letter denotes the protein conformation investigated: N for native peptide conformation, L for linear epitopes, and LN when both linear (in a larger cohort) and native (in a smaller subset) conformations were analyzed. D indicates that the antigen is no peptide. The letter following the first slash specifies the sample compartment: P for peripheral blood, C for cerebrospinal fluid, O when oligoclonal band reactivity alone was tested, and B when both blood and CSF were analyzed. The second slash introduces additional notations: OD for antigens showing reactivity in another disease, LF for low-frequency reactivity confined to a small subset of MS patients, CS for the absence of a control group (Case series), SP for findings restricted to progredient disease courses and SR for findings restricted to relapsing-remitting disease courses. The label R/OD identifies reviews addressing diseases other than MS.

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
