# Peer review of "The Elusive B Cell Antigen in Multiple Sclerosis: Time to Rethink CNS B Cell Functions"

_ijms, 2025, doi:10.3390/ijms262110771_

Round 1
Reviewer 1 Report
Comments and Suggestions for Authors
Summary
This review traces primary studies from 1981 through 2025 on MS B-cell antigen discovery. Using a PubMed search supported by AI tools and the authors’ own screening, the review identifies 26 immune targets. The authors discuss each target in detail and group them into three categories (non-specific, controversial, and provisional). They conclude that a primarily antibody-driven initiation is unlikely and instead propose a secondary B-cell response. Overall, the paper provides a valuable reference for MS B-cell targets, though several aspects could be further strengthened to enhance rigor and clarity.
Major Comments
- The literature search strategy appears too narrow. The authors are encouraged to broaden the coverage to include additional publication databases and preprints, and to expand the query with synonyms and related terms (e.g., B-cell antigen / autoantibody / epitope). A complete and reproducible search strategy should be reported in detail. AI tools could also be applied at the initial search stage to help identify relevant studies.
- Regarding the use of AI tools and selective reading, the authors should clarify whether any predefined rules, criteria, or prompts were used, to ensure transparency and reproducibility.
- The authors are supposed to organize the findings into a single comprehensive table (either newly created or integrated into Table 1). For each antigen, the table should include: (i) specificity status (MS-specific vs. also observed outside MS), (ii) key supporting evidence, (iii) key counter-evidence, and (iv) an overall classification (non-specific / controversial / provisional), all accompanied by explicit reference citations.
- Section 1.3 is not well aligned with the subsequent analysis. If this section is retained, the authors are encouraged to add a systematic MS vs NMOSD/MOGAD comparison across all candidate antigens. Otherwise, it is recommended to condense or reposition this section so that the background connects more directly with the Results.
- The discussion of methodological difficulties in Section 1.4 is too general to support the paper’s argument. The authors could use this section to explain why findings on individual antigens differ, explicitly linking assay designs and sample sources to variability in results. Adding these connections would make the introduction more relevant and informative for the subsequent analysis.
Minor Comments
- The manuscript is generally informative but would benefit from targeted language edits to improve fluency and reader flow. Long, multi-clause sentences and occasional jargon make key points harder to parse.
- The authors should standardize the spelling of GlialCAM throughout the manuscript (avoid mixing “GlialCam” and “GlialCAM”).
Reviewer 2 Report
Comments and Suggestions for Authors
The review is generally well-organized, current in many references, and maintains a balanced tone (the abstract and primary argument are distinct), but several specific issues hinder reproducibility, rigor, and clarity that must be resolved prior to acceptance; **Methods & reproducibility** — the search strategy is inadequately detailed: you mention a PubMed search using the terms `"Antibody target" AND "Multiple Sclerosis" NOT (Review[Publication Type])` with a timeframe extending to 2025, yet you need to specify the precise date(s) the search was conducted, the complete search strings (including quotation marks/MeSH, synonyms, and Boolean operators), any language restrictions, and whether additional databases (Embase, Scopus, Web of Science), preprint servers (medRxiv/bioRxiv), or conference abstracts were searched — this information is essential for reproducibility and preventing overlooked studies; include a formal PRISMA flowchart with specific numbers at each step (although you have Figure 1, it should adhere to PRISMA standards and provide reasons for exclusions). **Inclusion/exclusion & bias assessment** — you fail to specify clear inclusion/exclusion criteria (such as study design, minimum sample size, assay validation requirements, CSF versus serum, and cell-based versus denatured assays), nor do you conduct any systematic evaluations of study quality or risk of bias (for example, using the adapted Newcastle–Ottawa, QUADAS-2, or similar frameworks); without this detail, labeling antigens as “non-specific / controversial / provisional” becomes subjective — provide explicit operational definitions (for instance: “non-specific = reported in ≥2 independent cohorts including OND/HC; controversial = at least one high-quality study supporting and one disputing; provisional = single small study with no replication”) along with a table summarizing supporting versus refuting evidence, sample sizes, assay types, and whether the reactivity was in CSF or serum. **Use of AI tools** — you mention utilizing “Academic Perplexity” and “Research Rabbit”; in alignment with transparency standards, include the precise queries/prompts, the date and version of those tools, and how their outputs factored into the selection process (were potential candidates identified for subsequent manual validation?); outline any potential biases that may arise from relying on AI discovery tools. **Table 1 / data presentation** — Table 1 is informative but requires refinement: (i) each row should reference the original identifying study(ies) alongside subsequent replication/negative findings in distinct columns; (ii) unify the terminology for “Initial Identification Method” (cell-based assay / ELISA / phage display / Western blot / immunoprecipitation / microarray) and indicate if the antigen was in native conformation; (iii) the “Molecular Mimicry Status” column is ambiguous (entries like “0” vs “EBNA1” lack clarity) — change to more specific labels (e.g., “evidence for EBV-crossreactivity: yes/no; refs”); (iv) include a brief evidence score (1–5) to help readers quickly assess where evidence groups. **Key factual/interpretive points that need tempering or expansion** — when you state a “primarily antibody-mediated mechanism appears unlikely,” clarify to be more accurate (e.g., “current evidence does not support a singular, high-prevalence primary autoantibody explaining MS pathogenesis”) and clearly identify the strongest counterexamples (e.g., KIR4.1's initial report showing high prevalence followed by non-replication — examine assay differences and cohort overlaps instead of treating it as conclusive), and similarly approach Owens et al.'s PLP cell-binding findings as promising yet constrained by cohort size and lack of independent validation; provide effect sizes and cohort sizes when discussing prevalence. **Literature weighting & publication bias** — you mention that high-impact journals tend to publish positive results; consider including a concise methods section to quantify this (e.g., number of positive studies by journal impact or by laboratory), and address cohort overlap (same cohorts reported across various assays) which can create false replication; whenever feasible, indicate whether studies utilized independent cohorts. **Mechanistic and recommendation sections** — enhance the “alternative functions” section by providing specific experimental recommendations: (a) methodical cloning and assessment of OCB-derived recombinant antibodies featuring native antigen displays, (b) synchronized multi-center CSF/serum investigations employing standardized cell-based assays, (c) integrating repertoire sequencing with antigen-discovery (phage/CBA) and functional assessments, and (d) repeated sampling to differentiate primary from secondary responses — these practical next steps would enhance the translational impact of your review. **Minor / editorial issues** — amend typographical mistakes (e.g., “glykoprotein” → “glycoprotein” in the abbreviations), verify unclear abbreviations (e.g., “MaSp” seems ambiguous — likely means “Mass Spec”), standardize terminology (Nf vs Neurofascin; PLP/MBP/MOG consistently defined), and make sure that all table footnotes describe assay sensitivity/specificity where possible; additionally, since one author (MK) has various industry affiliations, briefly explain how COI was addressed in literature interpretation to reassure readers. **Concluding wording & strength of claims** — soften absolute language (change “unlikely” to “not strongly supported by current evidence” and include clear qualifications regarding small sample sizes, assay variability, and publication bias), and include a concise boxed summary outlining (i) antigens with most compelling evidence and their rationale, (ii) antigens needing urgent validation, and (iii) suggested standardized experimental methodologies; lastly, append a supplementary Excel/CSV evidence matrix detailing each referenced study, cohort size, assay type, findings, and DOI, enabling reviewers/readers to easily audit your evidence framework. If you wish, I can now (a) create specific content for the Methods section (exact language for the search strategy + PRISMA flow text), (b) transform Table 1 into a reproducible evidence matrix template for you to complete, and (c) offer proposed wording for the abstract/conclusion that more accurately represents the graded strength of evidence.
Comments on the Quality of English Language
Minor editing is required
